# Maternal hepatitis C virus infection and three adverse maternal outcomes in the United States

Robert B. Hood[1], William C. Miller[1], Abigail Shoben[2], Randall E. Harris[1], Alison H. Norris[1]*

1 Division of Epidemiology, College of Public Health, The Ohio State University, Columbus, OH, United States of America, 2 Division of Biostatistics, College of Public Health, The Ohio State University, Columbus, OH, United States of America

* norris.570@osu.edu

**Data Availability Statement:** All files are available from the CDC Vital Statistics Online Data Portal (year: 2017) https://www.cdc.gov/nchs/data_access/vitalstatsonline.htm.

## Abstract

### Background

Hepatitis Virus C (HCV) infection rates have trended upwards among pregnant people in the USA since 2009. Existing evidence about HCV infections and maternal outcomes is limited; therefore, we used birth certificate data to investigate the association between HCV infection and maternal health outcomes.

### Methods

We used the 2017 US birth certificate dataset (a cross-section of 1.4 million birth records) to assess the association between prevalent HCV infection and gestational diabetes, gestational hypertension, and eclampsia. Potential confounding variables included prenatal care, age, education, smoking, presence of sexually transmitted infections (STIs), body mass index (BMI), and weight gain during pregnancy. We restricted our analysis to only women with a first singleton pregnancy. Odds ratios were estimated by logistic regression models and separate models were tested for white and Black women.

### Results

Only 0.31% of the women in our sample were infected with HCV (n = 4412). In an unadjusted model, we observed a modest significant protective association between HCV infection and gestational diabetes (Odds ratio [OR]: 0.83; 95% CI: 0.76–0.96); but this was attenuated with adjustment for confounding variables (Adjusted odds ratio [AOR]: 0.88; 95% CI: 0.76, 1.02). There was no association between HCV and gestational hypertension (AOR: 1.03; 95% CI: 0.91, 1.16) or eclampsia (AOR: 1.15; 95% CI: 0.69, 1.93). Results from the race stratified models were similar to the non-stratified summary models.

### Conclusion

We observed no statistically significant associations between maternal HCV infection with maternal health outcomes. Although, our analysis did indicate that HCV may lower the risk

**Funding:** The author(s) received no specific funding for this work.

**Competing interests:** The authors have declared that no competing interests exist.

of gestational diabetes, this may be attributable to confounding. Studies utilizing more accurately measured HCV infection including those collecting type and timing of testing, and timing of infection are warranted to ensure HCV does not adversely impact maternal and/or fetal health. Particularly in the absence of recommended therapy for HCV during pregnancy.

## Introduction

Hepatitis C Virus (HCV) is the most common blood-borne infection in the United States (US) with an estimated 44,700 acute HCV infections occurring in 2017 [1, 2]. HCV can cause both acute and chronic infections [2]. Chronic HCV infections can lead to liver cirrhosis, hepatocellular carcinoma, and advanced liver disease necessitating a liver transplant [2–4]. Prior to 1992, the main mode of transmission was through contaminated blood products and injection drug use [3]. However, because blood products are routinely screened for HCV, injection drug use is now responsible for a majority of new HCV cases in the US [2, 3]. Driven by shared injection equipment during the ongoing opioid epidemic, from 2009 to 2017, HCV infections among pregnant women increased from 1.8 cases to 4.7 cases per 1000 live births [5, 6]. Given the increasing numbers of fetuses and expectant mothers exposed to HCV during pregnancy [5, 6], we examined the potential harm of HCV infection during pregnancy on maternal health.

During certain stages of pregnancy, the immune system is depressed, which allows for an increase in viral proliferation, which, in turn, may negatively impact the health of the pregnant person and the developing fetuses [5, 7–12]. Previous epidemiological investigations have demonstrated that infants born to pregnant people with HCV may be at increased risk for prematurity, low birth weight, and being small for gestational age [7, 8, 13]. Less is known, however, about the health of HCV infected people during pregnancy. Some epidemiological studies from Florida and Europe suggest that HCV infected women are at higher risk for developing gestational diabetes compared to uninfected women [8, 9]. However, another study, from Washington state, observed that the association between maternal HCV and gestational diabetes is modified by maternal weight gain [7]. Thus far, no association has been observed between maternal HCV infection and gestational hypertension or preeclampsia but few studies examine this association [8, 9]. The epidemiologic literature currently offers conflicting results and many of these studies lack generalizability to the larger US population.

Given that the US has one of the worst pregnancy-related mortality rates (17.2 maternal deaths per 1000 live births) among high-income countries [14]; understanding factors that may negatively impact the health of pregnant women in the US can provide useful insight for opportunities to improve maternal health. Furthermore, the increasing number of HCV infections during pregnancy in the US [5, 6] is concerning as the epidemiological literature is currently unclear as to if and how HCV can affect maternal health. To address this dearth in knowledge, we investigated the relationship between maternal HCV infection and maternal health, with a focus on gestational diabetes, gestational hypertension and eclampsia among women giving birth to their first singleton live birth in the US. Furthermore, we sought to understand the differing experiences of white and Black women given that Black women experience the consequences of structural racism, resulting in higher percentages of negative birth outcomes [15]. These analyses may inform healthcare providers about the potential risks that maternal HCV infection may have on maternal health as well as potential racial health disparities related to maternal HCV infection and maternal health in the US.

## Materials & methods

### Data

We obtained the 2017 US National Birth Certificate dataset from the National Center for Health Statistics (NCHS). This de-identified dataset contained most variables available in the 2003 edition of the US birth certificate for all US States, the District of Columbia, and the US territories. This dataset included more than 99% of all births that occurred in the US in 2017 and has several quality control measures to ensure it accurately captured birth data from state registries [16].

For this analysis, we excluded births from US territories because the healthcare systems in these areas likely differ from States and the District of Columbia in meaningful ways (n = 29,851) [17–19] (*Fig 1*). In an effort to isolate the effect of HCV on maternal health, we include records of women between the ages of 15–49 with a live birth following a first singleton pregnancy. Pregnancies resulting in multiple live births were excluded because these can have negative impacts on maternal health (n = 132,536). Birth certificate data do not include data on prior birth outcomes that can impact maternal morbidity, so we excluded multiparous women (n = 2,301,926). We restricted our analysis to women who were between 15 to 49 years old (n = 2,108) since births occurring in the youngest and oldest populations suffer from higher rates of complications [20]. For our analysis of gestational diabetes, we restricted our sample to women who did not have diabetes prior to pregnancy (n = 80,010). Similarly, for our analysis of gestational hypertension and eclampsia, we restricted our sample to women who did not report hypertension prior to pregnancy (n = 24,811). Finally, for our analysis we conducted a complete case analysis and restricted our analysis to women with complete data on the outcomes and covariates (n = 10,738 for gestational diabetes analysis; n = 77,202 for gestational hypertension analysis).

### Outcomes

We assessed three outcomes of interest: (1) gestational diabetes, (2) gestational hypertension, and (3) eclampsia. Each of these three conditions is listed as a checkbox on the birth certificate; the birth certificate information is typically filled out by hospital staff. On the birth certificate, gestational hypertension and preeclampsia are combined into one checkbox and will be referred to as gestational hypertension. We coded each outcome as 1 if the pregnant person was reported as having the condition and 0 if they did not. In each of the three outcomes not having the condition was the reference category.

### Covariates

Maternal HCV infection status, as reported on the birth certificate, was the main variable of interest. Having no reported HCV infection was the reference category. Maternal HCV infection status is abstracted from the person's medical records at the facility where the birth occurred. Specifically, on the form completed by medical staff at these facilities, they are asked: *"Infections present and/or treated during this pregnancy–present at the start of pregnancy or confirmed diagnosis during pregnancy with or without documentation of treatment. Documentation of treatment during this pregnancy is adequate if a definitive diagnosis is not present in the available record."* Facilities/reporters could indicate if gonorrhea, syphilis, chlamydia, hepatitis B, hepatitis C, or none of the above were present. The checkbox does not indicate if this is a current or previous infection, only that they tested positive for HCV but does not indicate which test was used (antibody versus PCR). Furthermore, these birth certificate data do not indicate if a person was screened for HCV.

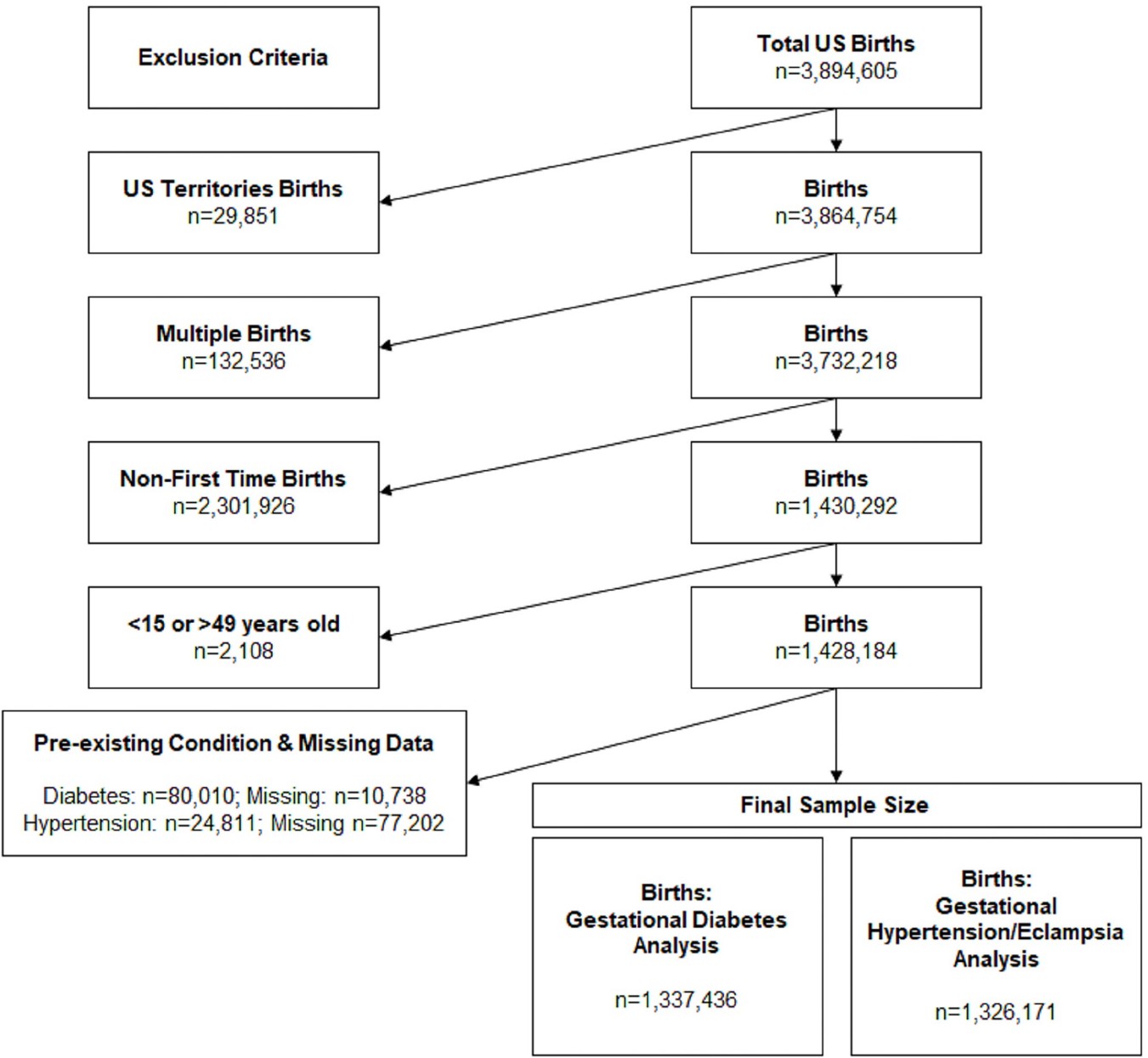

**Fig 1. Flowchart of study sample size of 2017 US birth certificate dataset applying exclusion criteria.**

We selected covariates by using a directed acyclic graph (DAG) and finding the minimal adjustment set for each outcome [21]. Our minimal adjustment set included prenatal care, maternal age, maternal education, maternal smoking status, the presence of sexually transmitted infections (STIs) besides HCV, body mass index (BMI) prior to pregnancy, and maternal weight gain during pregnancy.

We defined prenatal care using a continuous measure for the number of visits that occurred. We included a quadratic term for prenatal care due to the u-shaped relationship between number of prenatal care visits and these outcomes, where a small number of prenatal care visits may indicate inadequate utilization, and a large number of prenatal care visits may

reflect a health condition. We defined maternal age using a continuous measure which ranged from 15 to 49 years old. Again, we included a quadric term for maternal age given that maternal age has a non-linear relationship with maternal health during pregnancy. We used a nominal variable of maternal education with the following categories: some high school, high school diploma/GED, some college, bachelor's degree, advance degree, and unknown. Advance degree was used as a reference group for maternal education. Maternal smoking status was a categorical variable with four categories: non-smoker, smoked during one trimester, smoked during two trimesters, and smoked during three trimesters. Non-smoker was used as a reference category for maternal smoking. For presence of STIs, we used a dichotomous variable to indicate the presence or absence of a positive test for an STI. STIs tested for included Chlamydia, Gonorrhea, Hepatitis B Virus, and Syphilis. We used no STIs detected as the reference category. We defined BMI prior to pregnancy as an ordinal variable with the following categories: Underweight (less than 18.5), Normal weight (18.5 to 24.9), Overweight (25.0 to 29.9), Obesity I (30.0 to 34.9), Obesity II (35.0 to 39.9) and Extreme obesity III (40.0 or greater) [22]. We used normal weight as the reference category. Finally, we defined weight gain during pregnancy as an ordinal variable using guidelines from the Institute of Medicine and National Research Council with the following categories: less than the optimal weight gain, optimal weight gain, and more than the optimal weight gain. Weight gain categories were based on a women's starting BMI and how much weight they gained during pregnancy [23]. We used optimal weight gain as the reference category for this variable. Optimal weight gain for underweight, normal weight, overweight and obese women is: 28 to 40 pounds, 25 to 35 pounds, 15 to 25 pounds, and 11 to 20 pounds respectively [23].

## Statistical analysis

We used frequencies, medians, and interquartile ranges, to describe the 2017 birth cohort. We used a complete case analysis with imputed values from the NCHS. In our sample, we included single imputed values for maternal age ($<0.01\%$; n = 42) and maternal race (5.4%; n = 77,692). The NCHS' methods for single imputation have been described elsewhere. Briefly, when maternal age was missing, NCHS conducted a single imputation with the maternal age from a preceding birth record that had the same race and total birth order as the record with the missing maternal age value [16]. When maternal race was missing, NCHS either used a single imputation with the paternal race (if known) or with the maternal race of the preceding birth record (if paternal race was also missing) [16]. We analyzed the relationship between maternal HCV status and gestational diabetes, gestational hypertension, and eclampsia using unadjusted and adjusted logistic regression. To examine the differing experiences of white and Black women, we stratified by race with a model for white women and a model for Black women. Model stratification allowed us to compare the impact of structural racism on the collective experience of Black women against the collective experience of white women. All analysis took place in SAS (v 9.4, Cary NC).

## Ethical statement

This study was approved by the Institutional Review Board at The Ohio State University. Informed consent was not required for these analyses.

## Results

### Sample characteristics

In 2017, over 1.4 million women in the US gave birth for the first time and had singleton births (*Table 1*). The median age of the people was 27 years (Interquartile range (IQR): 9), most did

**Table 1. Demographic characteristics among women (15–49 years old) who had their first live singleton birth in the United States in 2017 by race.**

| Characteristic | All Women (n = 1,428,184) n (%) | White Women (n = 1,040,726) n (%) | Black Women (n = 214,143) n (%) |
|---|---|---|---|
| HCV Status | | | |
| Positive | 4412 (0.31) | 3909 (0.38) | 222 (0.10) |
| Negative | 1420651 (99.47) | 1034547 (99.41) | 213297 (99.60) |
| Missing | 3121 (0.22) | 2270 (0.22) | 624 (0.29) |
| Gestational Diabetes | | | |
| Yes | 78796 (5.52) | 53788 (5.17) | 9201 (4.30) |
| No | 1348174 (94.40) | 986069 (94.75) | 204709 (95.59) |
| Missing | 1214 (0.09) | 869 (0.08) | 233 (0.11) |
| Gestational Hypertension | | | |
| Yes | 117701 (8.24) | 87657 (8.42) | 20159 (9.41) |
| No | 1309269 (91.67) | 952200 (91.49) | 193751 (90.48) |
| Missing | 1214 (0.09) | 869 (0.08) | 233 (0.11) |
| Eclampsia | | | |
| Yes | 4882 (0.34) | 3217 (0.31) | 1030 (0.48) |
| No | 1422088 (99.57) | 1036640 (99.61) | 212880 (99.41) |
| Missing | 1214 (0.09) | 869 (0.08) | 233 (0.11) |
| Maternal Education | | | |
| < High school | 143661 (10.06) | 100275 (9.64) | 28929 (13.51) |
| High school / GED | 339751 (23.79) | 240390 (23.10) | 71012 (33.16) |
| Some college | 396319 (27.75) | 290866 (27.95) | 69143 (32.29) |
| Bachelor's degree | 331100 (23.18) | 252776 (24.29) | 28602 (13.36) |
| Advanced degree | 200229 (14.02) | 144988 (13.93) | 14238 (6.65) |
| Missing | 17124 (1.20) | 11431 (1.10) | 2219 (1.04) |
| Tobacco Use | | | |
| 1 Trimester | 17438 (1.22) | 14107 (1.36) | 1888 (0.88) |
| 2 Trimesters | 6731 (0.47) | 5602 (0.54) | 599 (0.28) |
| 3 Trimesters | 50575 (3.54) | 43817 (4.21) | 3589 (1.68) |
| Did not smoke | 1347764 (94.37) | 973218 (93.51) | 206990 (96.66) |
| Missing | 5676 (0.40) | 3982 (0.38) | 1077 (0.50) |
| Maternal BMI | | | |
| Underweight | 59071 (4.14) | 37914 (3.64) | 8810 (4.11) |
| Normal weight | 670093 (46.92) | 492234 (47.30) | 81160 (37.90) |
| Overweight/Obese | 666841 (46.69) | 489562 (47.04) | 117028 (54.65) |
| Missing | 32179 (2.25) | 21016 (2.02) | 7145 (3.34) |
| Sexually Transmitted Infections | | | |
| Yes | 37525 (2.63) | 19781 (1.90) | 12966 (6.05) |
| No | 1387538 (91.15) | 1018675 (97.88) | 200553 (93.65) |
| Missing | 3121 (0.22) | 2270 (0.22) | 624 (0.29) |
| | *Median (IQR)* | *Median (IQR)* | *Median (IQR)* |
| Maternal age | 27.0 (9.0) | 27.0 (9.0) | 24.0 (8.0) |
| Prenatal care visits | 12.0 (4.0) | 12.0 (4.0) | 11.0 (5.0) |

not smoke (94.4%; n = 1,347,764), and the majority of people were either normal weight (46.9%; n = 670,093) or overweight/obese (46.7%; n = 666,841).

Less than 1% of the sample had an indication of HCV infected status on the birth certificate (0.3%; n = 4,412). Approximately 5% of people experienced gestational diabetes (5.5%;

**Table 2. Association between maternal HCV infection and maternal health outcomes using logistic regression among women (15–49 years old) who gave birth to their first live singleton birth in the United States in 2017.**

| Outcome | All Women | | White Women [§] | | Black Women [§] | |
|---|---|---|---|---|---|---|
| | Unadjusted OR (95% CI) [†] | Adjusted [‡] OR (95% CI) [†] | Unadjusted OR (95% CI) [†] | Adjusted [‡] OR (95% CI) [†] | Unadjusted OR (95% CI) [†] | Adjusted [‡] OR (95% CI) [†] |
| Gestational Diabetes [¶] | 0.83 (0.72, 0.96) | 0.88 (0.76, 1.02) | 0.87 (0.74, 1.02) | 0.89 (0.76, 1.05) | 0.95 (0.47, 1.94) | 1.02 (0.50, 2.09) |
| Gestational Hypertension [¶] | 0.94 (0.83, 1.05) | 1.03 (0.92, 1.16) | 0.90 (0.80, 1.02) | 1.04 (0.91, 1.18) | 1.08 (0.67, 1.73) | 1.13 (0.70, 1.83) |
| Eclampsia [¶,††] | 1.20 (0.72, 2.00) | 1.15 (0.69, 1.93) | 0.99 (0.53, 1.84) | 0.93 (0.50, 1.75) | — | — |

[†] OR: Odds Ratio; 95% CI: 95% Confidence Interval

[‡] Logistic regression models were adjusted for prenatal care, maternal age, maternal education, maternal race, maternal smoking status, the presence of sexually transmitted infections, maternal body mass index prior to pregnancy, and weight gain during pregnancy

[§] Models were stratified by race to examine the experiences of White and Black women separately

[¶] Odds ratios describe the odds of developing the maternal outcome comparing women with an HCV infection to women without an HCV infection

[††] Estimates could not be obtained due to few cases of eclampsia among Black women

n = 78,796). Additionally, 8.2% of people experienced gestational hypertension during pregnancy (8.2%; n = 117,701). Very few people experienced eclampsia (0.3%; n = 4,882).

**Non-stratified models.** In an unadjusted model comparing all women with HCV infection to those without infection, a modest significant protective association was observed between HCV infection and gestational diabetes (Odds ratio (OR): 0.83; 95% Confidence Interval (CI): 0.72, 0.96) (*Table 2*). After adjustment, the association was attenuated (OR: 0.88; 95% CI: 0.76, 1.02). Associations of HCV infection with gestational hypertension and eclampsia fluctuated about 1.0 and were not statistically significant.

## Race stratified models

Approximately 1 million white women had first singleton births in 2017 and over 200,000 Black women had first singleton births in 2017 (*Table 1*). The median age for white women was 27 years (IQR: 9.0) and the median age for Black women was 24 years (IQR: 8.0). Among white women a nearly equal number were either normal weight (47.3%; n = 492,234) or overweight/obese (47.0%; n = 489,562). Among Black women, more women were overweight/obese (54.7%; n = 117,028) than normal weight (37.9%; n = 81,160). A higher percentage of white women (0.4%; n = 3,909) tested positive for HCV compared to Black women (0.1%; n = 222). However, more Black women tested positive for other STIs (6.05%; n = 12,966) compared to white women (1.90%; n = 19,781). Among the three maternal health outcomes, more white women had gestational diabetes (5.2% vs 4.3%) while more Black women had gestational hypertension (9.4% vs 8.4%) or eclampsia (0.5% vs. 0.3%).

Among white women, the inverse association between HCV infection and gestational diabetes observed for all women was not significant in the unadjusted model (OR: 0.87; 95% CI: 0.74, 1.02) or the adjusted models (OR: 0.89; 95% CI: 0.76, 1.05) (*Table 2*). Other adjusted odds ratios estimated separately for white women and Black women did not approach statistical significance. We did not estimate the odds ratio for maternal HCV infection and eclampsia among Black women due to small sample size.

## Discussion

In 2017, among the 1.4 million singleton first time births that occurred in the US, while less than one percent of pregnant women were positive for HCV (0.31%), this represents an increase in maternal HCV infections in the US as compared to earlier years [5, 6]. We observed

that maternal HCV infection had little association with various maternal health outcomes and that some of these associations may be modified by racial disparities, but these associations are not entirely clear.

Our findings indicate that maternal HCV infection may slightly lower the odds of developing gestational diabetes compared to HCV uninfected individuals. While the adjusted confidence interval is wide and includes the null value, a majority of the interval falls below the null value; thus, our result may be a function of the small number of individuals with HCV in our sample rather than an indication of a null association. Our findings are different from those of several studies which found that maternal HCV increased the risk for developing gestational diabetes [7–9]. However, one study that observed a higher prevalence of gestational diabetes in women infected with HCV compared to uninfected women did not account for confounding variables [8]. Furthermore, that analysis included several years of data before the 2003 birth certificate was instituted [8], which could have led to undercounting of HCV infections. In the other two studies, Pergam *et al.* only observed an increase in risk for gestational diabetes in HCV infected women who had excessive weight [7] and in the other, Reddick *et al.* utilized hospital discharge data from hospitals across 37 states [9]. In the current analyses, we included excessive weight gain as a continuous confounder rather than stratifying by it and the results are in line with what Pergam *et al.*, observed in their non-stratified analyses. In contrast to Reddick *et al.*, we utilized birth certificate data which covers all 50 states and included only a single year in which all states had fully instituted the 2003 birth certificate (which captures HCV data). Because of the national coverage and the complete use of the 2003 birth certificate, our dataset may therefore be more representative and likely had less undercounting of HCV cases. Additionally, Reddick *et al.* utilized a different adjustment set and included individuals with prior live births which we did not include due to potential confounding based on past pregnancy outcomes. Only one other study observed a similar trend to our findings, using all birth certificates from 2009 to 2017: Rossi *et al.* observed a lower prevalence of gestational diabetes among HCV infected women when compared to uninfected women. However, this association was not adjusted for confounding variables and is therefore not directly comparable [5]. It is not clear why maternal HCV infection might decrease the risk for gestational diabetes. One potential scenario is that HCV may increase the risk for development of Type II diabetes prior to pregnancy [24]. If a person were to develop Type II diabetes prior to pregnancy, they would no longer be classified as developing gestational diabetes during pregnancy, thus artificially decreasing the risk by removing individuals from the population at risk. However, without information about the timing of HCV infections and longitudinal studies of HCV among people of reproductive age, the potential pathway between maternal HCV infection and gestational diabetes remains unclear.

Our findings indicate that maternal HCV infection is not associated with gestational hypertension or eclampsia. The adjusted odds ratios for these outcomes did not approach statistical significance. Findings from other studies are mixed regarding the presence and/or direction of these associations. Connell *et al.*, using Florida birth certificates, observed that HCV infection was not associated with preeclampsia but observed that HCV infection was associated with a lower prevalence of gestational hypertension [8]. In contrast, a study of hospital discharge data found an association between HCV infection and lower risk of preeclampsia but this association was not statistically significant [9]. These results differ from the current study likely in part because of our use of a combined outcome measure that includes women with gestational hypertension or preeclampsia rather than separating these two outcomes. A recent analysis of birth certificate data (2009–2017), that also used a combined measure of gestational hypertension and preeclampsia, found no difference in the prevalence of these outcomes across HCV infection status [5]. However, this result was unadjusted and confounding cannot be ruled out.

Limited literature is available on the association between maternal HCV infection and eclampsia. Only a single study examined this relationship and found no difference in the prevalence of eclampsia between HCV infected and uninfected people [8]. In the current study, the number of cases was small and none of the odds ratios approached significance. Therefore, further study including longitudinal studies of maternal HCV and gestational hypertension and eclampsia are needed to clarify the relationships.

Our results indicate limited evidence for racial disparities in the association of HCV infection with gestational diabetes, gestational hypertension, and eclampsia. Generally, the results of the race stratified models followed the patterns from the overall models with a few exceptions. But even in the absence of notable racial disparities in these results, we posit that racism still impacts the relationship between maternal HCV infection and maternal health. Others have observed that pregnant Black women are screened for HCV at lower rates than pregnant white women [25]. The results of our study support this finding as Black women in our study had a much lower HCV infection prevalence while simultaneously had a higher proportion of other STIs when compared to white women. A lack of screening among Black women could lead to misclassification of HCV status which may mask potential associations between HCV and maternal health outcomes. More equitable screening practices need to be established to understand how racism and racial disparities impact maternal HCV infection and maternal health.

While this study has a large and nationally representative sample, we note several limitations. First, HCV can be diagnosed and confirmed by two tests: an antibody test and a PCR test. The antibody test indicates if an individual has *ever* had an infection whereas the PCR test determines if an individual has an *active* infection. From the birth certificate data, we cannot assess which test was performed, and who has an active infection. In an ideal study, we would have examined differences in maternal health between individuals who never had an HCV infection, individuals who had an HCV infection but cleared it or received treatment, and individuals who had an active HCV infection. Both active and prior HCV infections could carry different risk for maternal health. If an individual has an active infection, their liver is likely being continuously damaged, and individuals with a prior HCV infection could have extensive liver damage depending on the length and intensity of the infection. Therefore, future studies investigating maternal HCV infection should consider delineating between active and prior infections. Additionally, HCV is a largely asymptomatic disease for several decades of infection, and until recently, HCV screening was risk-based [26, 27]. Several studies have found that risk-based screening misses many individuals with HCV [4, 28–32], and a better screening protocol would include screening all pregnant women for HCV. Because of these two limitations, there is potential misclassification of HCV status in birth certificate data. This bias is likely non-differential, which may bias results towards the null. In other words, having failed to include HCV infected individuals in the correct category, we may see no statistically significant association even when one truly exists. Another limitation is the small number of gestational diabetes, gestational hypertension, and eclampsia outcomes observed. Because of the small number of events in the dataset, statistical significance could not be established in many models. However, we present these findings because this dataset is the largest available for births in the US and provides the best information on maternal outcomes. Finally, these results stem from a cross-sectional study, and thus temporality and timing of HCV infection could not be established. These results however do generate useful hypotheses that should be investigated further with longitudinal studies.

Over the past few years maternal HCV infections have been rising in the US. Our study used US birth certificate data to investigate the association between maternal HCV infection and three prevalent adverse maternal outcomes, gestational diabetes, gestational hypertension,

and eclampsia. We did not observe an association between maternal HCV infection and any of these outcomes, which may indicate that other than continued monitoring of the infection, changes are not currently needed for HCV guidelines during pregnancy for maternal health outcomes. However due to limitations in the way maternal HCV information is collected in birth certificate data, longitudinal studies with more adequate testing information will be necessary to fully explore if there is a null association between maternal HCV infection and adverse maternal health outcomes.

## Author Contributions

**Conceptualization:** Robert B. Hood, Abigail Shoben, Alison H. Norris.

**Methodology:** Robert B. Hood, Abigail Shoben, Alison H. Norris.

**Writing – original draft:** Robert B. Hood.

**Writing – review & editing:** Robert B. Hood, William C. Miller, Abigail Shoben, Randall E. Harris, Alison H. Norris.

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
