## [Decision Letter · Decision Letter 0]

28 Mar 2023

PONE-D-23-03216Maternal Hepatitis C virus infection and adverse maternal outcomes in the United StatesPLOS ONE

Dear Dr. Norris,

Thank you for submitting your manuscript to PLOS ONE. After careful consideration, we feel that it has merit but does not fully meet PLOS ONE’s publication criteria as it currently stands. Therefore, we invite you to submit a revised version of the manuscript that addresses the points raised during the review process.

We look forward to receiving your revised manuscript.

Kind regards,

Gbenga Olorunfemi, MBBS,MSC,FMCOG,FWASC

Academic Editor

PLOS ONE

Journal Requirements:

2. Please note that in order to use the direct billing option the corresponding author must be affiliated with the chosen institute. Please either amend your manuscript to change the affiliation or corresponding author, or email us at plosone@plos.org with a request to remove this option.

Reviewers' comments:

Reviewer's Responses to Questions

**Comments to the Author**

1. Is the manuscript technically sound, and do the data support the conclusions?

Reviewer #1: Partly

Reviewer #2: Yes

2. Has the statistical analysis been performed appropriately and rigorously? 

Reviewer #1: Yes

Reviewer #2: Yes

3. Have the authors made all data underlying the findings in their manuscript fully available?

Reviewer #1: Yes

Reviewer #2: No

4. Is the manuscript presented in an intelligible fashion and written in standard English?

Reviewer #1: Yes

Reviewer #2: Yes

5. Review Comments to the Author

Reviewer #1: This is an interesting study aimed at investigating the association between HCV infection and maternal health outcomes.

There are a number of issues that need to be addressed.

1. What is the study design? This is not apparent in the abstract.

2. The authors used a number of abbreviations that were not defined before use. Example: STIs, BMI

3. The authors stated: Only 0.31% of the women in our sample were infected with HCV (n=4412). How was the HCV confirmed? Was PCR technology employed in the confirmation of diagnosis?

4. The authors should beef up the justification for the study in the introduction.

5. How was the sample size determined in this study?

6.What is the clinical implications of the study findings?

7.The conclusion of the manuscript is not apparent.

8. What is the referencing style used by authors in the manuscript? It appears poorly done.

9. What is the flow chart of the participants in the study?

Reviewer #2: Thank you for asking me to review this manuscript on the adverse pregnancy outcome of Hepatitis C virus in the united states of america. The study was well written. However, I have some concern as outlined below

Abstract: Results: Authors wrote " In univariate models, we observed a modest significant inverse association between HCV infection and gestational diabetes (Odds ratio [OR]:0.83; 95% CI: 0.76-0.96);"

This inverse association in usual in the interpretation of odds ratio. Authors should revise

Line 161 . Authors stated " When maternal age is missing is it imputed with the age from a preceding record that has the same race and total birth order as the record with the missing value" The way authors handled missing data should be better described for the sake of reproducibility.

line 179: what is this :?? ',093'

Results: Tables comparing the proportion of GDM, hypertension, eclampsia among HCV positive and HCV negative is very germaine to this study.

presenting only three adverse outcome cannot be representative of "all adverse outcomes" Authors should adjust the title of the manuscript to reflect that only three adverse outcomes were studied

Table 1 should have test statistics and P-value. This should be described in the methods section to a reasonable extent

Line 189: "main analysis". This is not a good description of the results under this section. Authors should change "main analysis" to other phrases

line 191 "modest significant inverse association was obs" authors should rephrase

References: Authors should review the references to strictly conform to Vancouver reference style

Thank you

6. PLOS authors have the option to publish the peer review history of their article (what does this mean?). If published, this will include your full peer review and any attached files.

Reviewer #1: **Yes: **George Eleje

Reviewer #2: No

---

## [Author Response · Author response to Decision Letter 0]

19 Jul 2023

Reviewer #1:

1. What is the study design? This is not apparent in the abstract.

We used a cross-sectional study design. We have clarified this in the text of the abstract. The updated text reads (Lines 35-36): “We used the 2017 US birth certificate dataset (a cross-section of 1.4 million birth records) …”

2. The authors used a number of abbreviations that were not defined before use. Example: STIs, BMI

We have corrected this issue throughout the manuscript.

3. The authors stated: Only 0.31% of the women in our sample were infected with HCV (n=4412). How was the HCV confirmed? Was PCR technology employed in the confirmation of diagnosis?

In the US, the clinical guidelines suggest that HCV is screened using a two-test system. Individuals are typically screened first with an antibody test, which describes the presence or absence of HCV. However, the antibody test does not indicate if the HCV infection is an active or previous infection. Individuals with a positive HCV antibody test are then tested with a PCR test which will indicate if it is an active infection. 

Our study utilizes secondary data collected as a part of the US birth certificate. The HCV from this dataset indicates if a positive diagnosis was present in the patient’s medical record but does not indicate which test was conducted. We have stressed this limitation of this variable in the following lines of text (Lines 325-328): “First, HCV can be diagnosed and confirmed with two tests: an antibody test and a PCR test. The antibody test indicates if an individual has ever had an infection whereas the PCR test determines if an individual has an active infection. From the birth certificate data, we cannot assess which test was performed, and who has an active infection. …”

We have also clarified this point in the methods section. The updated text reads (Lines 145-147): “… The checkbox does not indicate if this is a current or previous infection, only that they tested positive for HCV but does not indicate which test was used (antibody versus PCR).”

4. The authors should beef up the justification for the study in the introduction.

We have added additional text to strengthen the rationale for this study. The updated text reads (Lines 87-92): “Given that the US has one of the worst pregnancy-related mortality rates (17.2 maternal deaths per 1000 live births) among high-income countries; understanding factors that may negatively impact the health of pregnant women in the US can provide useful insight for opportunities to improve maternal health. Furthermore, the increasing number of HCV infections during pregnancy in the US is concerning as the epidemiological literature is currently unclear as to if and how HCV can affect maternal health. To address this dearth in knowledge …”

5. How was the sample size determined in this study?

Our study was a secondary data analysis of existing US birth certificate data, because of this we did not conduct a formal sample size calculation since we would not have been able to recruit additional participants. Instead, we relied on the large sample size of the birth certificate dataset (approximately 1.4 million included in our study). 

However, based on a post-hoc sample size calculation assuming an alpha=0.05 and power=80% with an odds ratio from previous reports of 0.82. We would require a sample of approximately 15,695 participants. Thus, our sample size is more than adequate to detect a statistically significant difference.

6. What is the clinical implications of the study findings?

At this time, our study does not suggest a need to change clinical practice. However, we caution against over interpretation of these results. The updated text reads (Lines 355-361): “We did not observe an association between maternal HCV infection and any of these outcomes, which may indicate that, other than continued monitoring of the infection, changes are not currently needed for HCV guidelines during pregnancy for maternal health outcomes. However due to limitations in the way maternal HCV information is collected in birth certificate data, longitudinal studies with more adequate testing information will be necessary to fully explore if there is a null association between maternal HCV infection and adverse maternal health outcomes.”

 

7. The conclusion of the manuscript is not apparent.

We have clarified the conclusions of this manuscript. The updated text reads (Lines 351-361): “Over the past few years maternal HCV infections have been rising in the US. Our study used US birth certificate data to investigate the association between maternal HCV infection and three prevalent adverse maternal outcomes, gestational diabetes, gestational hypertension, and eclampsia. We did not observe an association between maternal HCV infection and any of these outcomes. However, due to limitations in the way maternal HCV infection information is collected in birth certificate data, longitudinal studies with more adequate testing information will be necessary to fully explore if there is a null association between maternal HCV infection and adverse maternal health outcomes.”

8. What is the referencing style used by authors in the manuscript? It appears poorly done.

We have corrected the errors with the Vancouver style citation format per the journal guidelines.

9. What is the flow chart of the participants in the study?

We have added Figure 1, which provides a flowchart of participants after applying our exclusion criteria. We have also added the number of participants excluded to the text.

Reviewer #2:

10. Abstract: Results: Authors wrote " In univariate models, we observed a modest significant inverse association between HCV infection and gestational diabetes (Odds ratio [OR]:0.83; 95% CI: 0.76-0.96);" This inverse association in usual in the interpretation of odds ratio. Authors should revise.

We have clarified this statement. The updated text reads (Lines 43-44): “In an unadjusted model, we observed a modest significant protective association between HCV infection and gestational diabetes…”

 

11. Line 161. Authors stated, " When maternal age is missing is it imputed with the age from a preceding record that has the same race and total birth order as the record with the missing value" The way authors handled missing data should be better described for the sake of reproducibility.

We have clarified how this single imputation method that was conducted by NCHS, the division of the CDC that maintains these data. We have also added a citation for individuals who wish to learn more about NCHS’ methods. The updated text reads (Lines 182-189): “We used a complete case analysis with imputed values from the NCHS. In our sample, we included single imputed values for maternal age (<0.01%; n=42) and maternal race (5.44%; n=77,692). The NCHS’ methods for single imputation have been described elsewhere. Briefly, when maternal age was missing, NCHS conducted a single imputation with the maternal age from a preceding birth record that had the same race and total birth order as the record with the missing maternal age value. When maternal race was missing, NCHS either used a single imputation with the paternal race (if known) or with the maternal race of the preceding birth record (if paternal was also missing).”

12. Line 179: what is this :?? ',093'

We have clarified that these numbers represent that 670,093 of individuals were normal weight and 666,841 individuals were overweight/obese in this sample.

13. Results: Tables comparing the proportion of GDM, hypertension, eclampsia among HCV positive and HCV negative is very germaine to this study.

We agree with the reviewer that this would be of interest. However, we would prefer not to include this table for two reasons. First, our unadjusted regression models provide an estimate of ratio of those who have these specific conditions by HCV infection status. Therefore, providing these numbers would only provide duplicate information and could lead readers to over interpret these proportions prior to any control for confounding. Second, due to our DUA with NCHS, we were only able to retain these data for one year, after which we had to destroy these data. We could request these data again, but would require an additional six months for the review and approval process. 

14. Presenting only three adverse outcome cannot be representative of "all adverse outcomes" Authors should adjust the title of the manuscript to reflect that only three adverse outcomes were studied.

We have changed the title to: “Maternal Hepatitis C virus infection and three adverse maternal outcomes in the United States.” 

 

15. Table 1 should have test statistics and P-value. This should be described in the methods section to a reasonable extent.

Because our sample includes over one million observations, we opted not to include statistical tests in Table 1. The sample size would cause p-values to shrink giving the false sense of precision for differences that would likely not be relevant either at the population level or the clinical level.

16. Line 189: "main analysis". This is not a good description of the results under this section. Authors should change "main analysis" to other phrases

We have changed “Main Analysis” to “Non-Stratified Models.”

17. Line 191 "modest significant inverse association was obs" authors should rephrase

We have rephased this statement. The updated text reads (Lines 214-216): “In an unadjusted model comparing all women with HCV infection to those without infection, a model significant protective association was observed between HCV infection and gestational diabetes…”

18. References: Authors should review the references to strictly conform to Vancouver reference style

We have corrected the errors with the Vancouver style citation format per the journal guidelines.

---

## [Decision Letter · Decision Letter 1]

11 Sep 2023

Maternal Hepatitis C virus infection and three adverse maternal outcomes in the United States

PONE-D-23-03216R1

Dear Dr. Norris,

We’re pleased to inform you that your manuscript has been judged scientifically suitable for publication and will be formally accepted for publication once it meets all outstanding technical requirements.

Kind regards,

Gbenga Olorunfemi, MBBS,MSC,FMCOG,FWASC

Academic Editor

PLOS ONE
---

## [Editor Report · Acceptance letter]

6 Oct 2023

PONE-D-23-03216R1 

Maternal Hepatitis C virus infection and three adverse maternal outcomes in the United States 

Dear Dr. Norris:

I'm pleased to inform you that your manuscript has been deemed suitable for publication in PLOS ONE. Congratulations! Your manuscript is now with our production department. 

Kind regards, 

on behalf of

Dr. Gbenga Olorunfemi 

Academic Editor

PLOS ONE